# Graph Neural Networks Do Not Always Oversmooth

**Bastian Epping**[1]**, Alexandre René**[1]**, Moritz Helias**[2,3]**, Michael T. Schaub**[1]

[1]RWTH Aachen University, Aachen, Germany
[2]Department of Physics, RWTH Aachen University, Aachen, Germany
[3]Institute for Advanced Simulation (IAS-6), Computational and Systems Neuroscience,
Jülich Research Centre, Jülich, Germany
`epping@cs.rwth-aachen.de, rene@cs.rwth-aachen.de,`
`m.helias@fz-juelich.de, schaub@cs.rwth-aachen.de`

## Abstract

Graph neural networks (GNNs) have emerged as powerful tools for processing relational data in applications. However, GNNs suffer from the problem of over-smoothing, the property that features of all nodes exponentially converge to the same vector over layers, prohibiting the design of deep GNNs. In this work we study oversmoothing in graph convolutional networks (GCNs) by using their Gaussian process (GP) equivalence in the limit of infinitely many hidden features. By generalizing methods from conventional deep neural networks (DNNs), we can describe the distribution of features at the output layer of deep GCNs in terms of a GP: as expected, we find that typical parameter choices from the literature lead to oversmoothing. The theory, however, allows us to identify a new, non-oversmoothing phase: if the initial weights of the network have sufficiently large variance, GCNs *do not* oversmooth, and node features remain informative even at large depth. We demonstrate the validity of this prediction in finite-size GCNs by training a linear classifier on their output. Moreover, using the linearization of the GCN GP, we generalize the concept of propagation depth of information from DNNs to GCNs. This propagation depth diverges at the transition between the oversmoothing and non-oversmoothing phase. We test the predictions of our approach and find good agreement with finite-size GCNs. Initializing GCNs near the transition to the non-oversmoothing phase, we obtain networks which are both deep and expressive.

## 1 Introduction

Graph neural networks (GNNs) reach state of the art performance in diverse application domains with relational data that can be represented on a graph, transferring the success of machine learning to data on graphs [47, 12, 23, 7]. Despite their good performance, GNNs come with the limitation of *oversmoothing*, a phenomenon where node features converge to the same state exponentially fast for increasing depth [35, 45, 30, 2]. Consequently, only shallow networks are used in practice [19, 1]. In contrast, it is known that the depth (i.e. the number of layers) is key to the success of deep neural networks (DNNs) [32, 33]. While for conventional DNNs shallow networks are proven to be highly expressive [6], in practice deep networks are much easier to train and are thus the commonly used architectures [36]. Furthermore, in most GNN architectures each layer only exchanges information between neighboring nodes. Deep GNNs are therefore necessary to exchange information between nodes that are far apart in the graph [9]. In this study, we investigate oversmoothing in graph convolutional networks (GCNs) [19].

To study the effect of depth, we consider the propagation of features through the network: given some input $x_\alpha^{(0)}$, each intermediate layer $l$ produces features $x_\alpha^{(l)}$ which are fed to the next layer. We

38th Conference on Neural Information Processing Systems (NeurIPS 2024).

follow the same approach that has successfully been employed in previous work to design trainable DNNs [37]: consider two nearly identical inputs $\boldsymbol{x}_\alpha^{(0)}$ and $\boldsymbol{x}_\beta^{(0)}$ and ask whether the intermediate features $\boldsymbol{x}_\alpha^{(l)}$ and $\boldsymbol{x}_\beta^{(l)}$ become more or less similar as a function of depth $l$. In the former case, the inputs may eventually become indistinguishable. In the latter case, the inputs become less similar over layers: the distance between them increases over layers [32, 37] until eventually it is bounded by the non-linearities of the network. The distance then typically converges to a fixed value determined by the network architecture, independent of the inputs.

One can therefore identify two phases: One says that a network is *regular* if two inputs eventually converge to the same value as function of $l$; conversely, one says that a network is *chaotic* if two inputs remain distinct for all depths [24]. Neither phase is ideal for training deep networks since in both cases all the information from the inputs is eventually lost; the typical depth at which this happens is called the *information propagation depth*. However, this propagation depth diverges at the transition between the two phases, allowing information – in principle – to propagate infinitely deep into the network. While these results are calculated in the limit of infinitely many features in each hidden layer, the information propagation depth has been found to be a good indicator of how deep a network can be trained [37]. A usual approach for conventional DNNs is thus to initialize them at the transition to chaos. Indeed, Schoenholz et al. [37] were able to use this approach to train fully-connected, feedforward networks with hundreds of layers. Similar methods have recently been adapted to the study of transformers, successfully predicting the best hyperparameters for training [5].

In this work we address the oversmoothing problem of GCNs by extending the framework described above in the limit of infinite feature dimensions from DNNs to GCNs: here the two different inputs $\boldsymbol{x}_\alpha^{(0)}$ and $\boldsymbol{x}_\beta^{(0)}$ correspond to the input features on two nodes, labeled $\alpha$ and $\beta$. The mixing of information across different nodes implies that output features on node $\alpha$ depend on input features on node $\beta$ and vice versa. Thus it is not possible to look at the distance of $\boldsymbol{x}_\alpha^{(l)}$ and $\boldsymbol{x}_\beta^{(l)}$ independently for each pair $\alpha$ and $\beta$ as in the DNN case. Rather, one has to solve for the distance between each distinct pair of nodes in the graph simultaneously: the one dimensional problem for DNNs thus becomes a multidimensional problem for GCNs. However, by linearizing the multidimensional GCN dynamics, we can generalize the notion of information propagation depth to GCNs: instead of being a single value, we find that a given GCN architecture comes with a set of potentially different information propagation depths, each corresponding to one eigendirection of the linearized dynamics of the system.

This approach allows us to extend the concept of a regular and a chaotic phase to GCNs: in the regular phase, which describes most of the GCNs studied in the current literature, distances between node features shrink over layers and exponentially attain the same value. We therefore call this the oversmoothing phase. On the other hand, if one increases the variance of the weights at initialization, it is possible to transition into the chaotic phase. In this phase, distances between node features converge to a fixed but finite distance at infinite depth. The convergence point is fully determined by the underlying graph structure and the hyperparameters of the GCN and may differ for different pairs of nodes. GCNs initialized in this phase thus do not suffer from oversmoothing. We find that the convergence point is informative about the topology of the underlying graph and may be used for node classification with GCNs of more than $1,000$ layers. Near the transition point, GCNs at large depth offer a trade-off between feature information and information contained in the neighborhood relation of the graph. We test the predictions of this theory and find good agreement in comparison to finite-size GCNs applied to the contextual stochastic block model [8]. On the citation network Cora [19] we reach the performance reported in the original work by Kipf and Welling [19] beyond $100$ layers. Our approach applies to graphs with arbitrary topologies, depths, and non-linearities.

## 2   Related Work

Oversmoothing is a well-known challenge within the GNN literature [21, 35]. On the theoretical side, rigorous techniques have been used to prove that oversmoothing is inevitable. The authors in [30] show that GCNs with the ReLU non-linearity exponentially lose expressive power and only carry information of the node degree in the infinite layer limit. While the authors notice that their upper bound for oversmoothing does not hold for large singular values of the weight matrices, they do not

identify a non-oversmoothing phase in their model. These results have been extended in [2] to handle non-linearities different from ReLU. Also graph attention networks have been proven to oversmooth inevitably [45]. Their proof, however, makes assumptions on the weight matrices which, as we will show, exclude networks in the chaotic, non-oversmoothing phase.

On the applied side, a variety of heuristics have been developed to mitigate oversmoothing [49, 4, 3, 22, 40, 15]. E.g., the authors in [49] introduce a normalization layer which can be added to a variety of deep GNNs to make them trainable. Another approach is to introduce residual connections and identity mappings, directly feeding the input to layers deep in the network [4, 46]. Other studies suggest to train GNNs to a limited number of layers to obtain the optimal amount of smoothing [18, 46]. The recent review [17] proposes a unified view to order existing heuristics and guide further research. While these heuristics improve performance at large depths, they also add to the complexity of the model and impose design choices. Our approach, on the other hand, explains why increasing the weight variance at initialization is sufficient to prevent oversmoothing.

## 3 Background

### 3.1 Network architecture

In this paper we study a standard graph convolutional network (GCN) architecture [19] with an input feature matrix $\boldsymbol{X}^{(0)} \in \mathbb{R}^{N \times d_0}$, where $N$ is the number of nodes in the graph and $d_0$ the number of input features. Bold symbols throughout represent vector or matrix quantities in feature space. The structure of the graph is represented by a shift operator $A \in \mathbb{R}^{N \times N}$. We write the features of the network's $l$-th layer as $\boldsymbol{X}^{(l)} \in \mathbb{R}^{N \times d_l}$; they are computed recursively as

$$\boldsymbol{X}^{(l)} = \phi(A\boldsymbol{X}^{(l-1)}\boldsymbol{W}^{(l)\top} + 1\boldsymbol{b}^{(l)\top}), \tag{1}$$

with $\phi$ an elementwise non-linear activation function, $\boldsymbol{b}^{(l)}$ an optional bias term, weight matrices $\boldsymbol{W}^{(l)} \in \mathbb{R}^{d_l \times d_{l-1}}$, and $1 \in \mathbb{R}^N$ a vector of all ones. We note that many GNN architectures studied in the literature are unbiased, which can be recovered by setting $\boldsymbol{b}^{(l)}$ to zero. We use a noisy linear readout, so that the output of the network is given by

$$\boldsymbol{Y} = A\boldsymbol{X}^{(L)}\boldsymbol{W}^{(L+1)\top} + 1\boldsymbol{b}^{(L+1)\top} + \boldsymbol{\epsilon}, \tag{2}$$

with $\boldsymbol{\epsilon} \in \mathbb{R}^{N \times d_{L+1}}$ being independent Gaussian random variables: $\epsilon_{\alpha,i} \overset{\text{i.i.d.}}{\sim} \mathcal{N}(0, \sigma_{ro}^2)$. The readout noise $\boldsymbol{\epsilon}$ is included both to promote robust outputs and to prevent numerical issues in the matrix inversion in Equations (11) and (12). We use $d_{L+1}$ to denote the dimension of outputs.

For the following it will be useful to consider the activity of individual nodes. To avoid ambiguity in the indexing, we use lower Greek indices for nodes and upper Latin indices for layers. We thus rewrite (1) as

$$\boldsymbol{x}_\alpha^{(l)} = \phi(\boldsymbol{h}_\alpha^{(l)}), \tag{3}$$

$$\boldsymbol{h}_\alpha^{(l)} = \sum_\beta A_{\alpha\beta}\boldsymbol{W}^{(l)}\boldsymbol{x}_\beta^{(l-1)} + \boldsymbol{b}^{(l)}, \tag{4}$$

$$\boldsymbol{y}_\alpha = \boldsymbol{h}_\alpha^{(L+1)} + \boldsymbol{\epsilon}_\alpha, \tag{5}$$

where $\boldsymbol{x}_\alpha^{(l)} \in \mathbb{R}^{d_l}$ is the feature vector of node $\alpha$ in layer $l$ and $\boldsymbol{y}_\alpha \in \mathbb{R}^{d_{L+1}}$ the network output for node $\alpha$. The values $\boldsymbol{h}_\alpha^{(l)} \in \mathbb{R}^{d_l}$ are linear functions of the features $\boldsymbol{x}_\beta^{(l-1)}$ and represent the input to the activation functions; we therefore refer to them as preactivations. The non-linearity $\phi(x)$ is applied elementwise to the preactivations $\boldsymbol{h}_\alpha^{(l)}$. While we leave $\phi(x)$ general for the development of the theory, we use $\phi(x) = \text{erf}(\frac{\sqrt{\pi}}{2}x)$ for the experiments in Section 4; this choice allows us to carry out certain integrals analytically. The scaling factor in the erf is chosen such that $\frac{\partial\phi}{\partial x}(0) = 1$. We use independent and identical Gaussian priors for all weight matrices and biases, $W_{ij}^{(l)} \overset{\text{i.i.d.}}{\sim} \mathcal{N}(0, \frac{\sigma_w^2}{d_l})$ and $b_i^{(l)} \overset{\text{i.i.d.}}{\sim} \mathcal{N}(0, \sigma_b^2)$ with $W_{ij}^{(l)}$ and $b_i^{(l)}$ being the matrix or vector entries of $\boldsymbol{W}^{(l)}$ and $\boldsymbol{b}^{(l)}$, respectively. As a shift operator, we choose

$$A = \mathbb{I} - \frac{g}{d_{\max}}(D - \mathcal{A}), \tag{6}$$

where $\mathcal{A}$ is the adjacency matrix, $\mathbb{I}$ the identity in $\mathbb{R}^{N \times N}$, $D_{\alpha\beta} = \delta_{\alpha\beta} \sum_{\gamma} \mathcal{A}_{\alpha\gamma}$ is the degree matrix and $d_{\max}$ is the maximal degree. The parameter $g \in (0, 1)$ allows us to weigh the off-diagonal elements compared to the diagonal ones. By construction the shift operator is row-stochastic, which means that it has constant sums over columns $\sum_{\beta} A_{\alpha\beta} = 1$. We will make use of this property in our analysis in Section 4.2. The generalization to non-stochastic shift operators will be shortly addressed later.

## 3.2 Gaussian process equivalence of GCNs

In a classic machine learning setting, such as classification, one draws random initial values for all parameters and subsequently trains the parameters by optimizing the weights and biases to minimize a loss function. This learned parameter set is then used to classify unlabeled inputs. In this paper we take a Bayesian point of view in which the network parameters are random variables, inducing a probability distribution over outputs which becomes Gaussian in the limit of infinitely many features. Thus infinitely wide neural networks are equivalent to Gaussian processes (GPs) [27, 43, 34]. In the study of DNNs this is a standard approach, yielding results which empirically hold also for finite-size networks trained with gradient descent [37].

In previous work [28, 14, 29] it has been shown that also the GCN architecture described in Section 3.1 is equivalent to a GP in the limit of infinite feature space dimensions, $d_l \to \infty$ for all hidden layers $l = 1, \ldots, L$, while input and readout layer still have tunable, finitely many features. In the GP description, all features are Gaussian random variables with zero mean and identical prior variance in each feature dimension. The description of the GCN thus reduces to a multivariate normal,

$$H^{(l)} \sim \mathcal{N}(0, K^{(l)}), \tag{7}$$

where $H^{(l)}$ is the vector of hidden node features of layer $l$, $H^{(l)} = (h_0^{(l)}, h_1^{(l)}, \ldots, h_N^{(l)})^\top$ under the prior distribution of weights and biases. The covariance matrices $K^{(l)} \in \mathbb{R}^{N \times N}$ are determined recursively: knowing that the $h_\alpha^{(l)}$ follow a zero-mean Gaussian with covariance $\langle h_\delta^{(l)} h_\gamma^{(l)} \rangle = K_{\delta\gamma}^{(l)}$, we define

$$C_{\gamma\delta}^{(l)} = \left\langle \phi\left(h_\gamma^{(l)}\right) \phi\left(h_\delta^{(l)}\right) \right\rangle_{h_\gamma^{(l)}, h_\delta^{(l)}}. \tag{8}$$

For simplicity we use $\phi(x) = \mathrm{erf}(\frac{\sqrt{\pi}}{2} x)$ for which Equation (8) can be evaluated analytically; see Appendix A for details. It follows from (3) that

$$K_{\alpha\beta}^{(l+1)} = \sigma_b^2 + \sigma_w^2 \sum_{\gamma, \delta} A_{\alpha\gamma} A_{\beta\delta} C_{\gamma\delta}^{(l)}, \tag{9}$$

as shown in [28, 29].

In a semi-supervised node classification setting, we split the underlying graph into $N^{\text{test}}$ unlabeled test nodes and $N^{\text{train}}$ labeled training nodes ($N^{\text{test}} + N^{\text{train}} = N$); we correspondingly split the output random variable $Y_i \in \mathbb{R}^N$ for output dimension $i$ into $Y_i^\star \in \mathbb{R}^{N^{\text{test}}}$ and $Y_i^D \in \mathbb{R}^{N^{\text{train}}}$. Features on the test nodes are predicted by conditioning on the values of the training nodes: $p(Y_i^* = y_i^* \mid Y_i^D = y_i^D)$. This leads to the following posterior for the unobserved labels (see [34, 20] for details):

$$Y_i^\star \sim \mathcal{N}(m_i^{\text{GP}}, K^{\text{GP}}), \tag{10}$$

$$m_i^{\text{GP}} = K_{\star D}^{(L+1)} (K_{DD}^{(L+1)} + \mathbb{I}\sigma_{ro}^2)^{-1} Y_i^D, \tag{11}$$

$$K^{\text{GP}} = K_{\star\star}^{(L+1)} - K_{\star D}^{(L+1)} (K_{DD}^{(L+1)} + \mathbb{I}\sigma_{ro}^2)^{-1} (K_{\star D}^{(L+1)})^\top. \tag{12}$$

Here the $\star$ and $D$ indices represent test and training data, respectively, i.e. $K_{DD} \in \mathbb{R}^{N^{\text{train}} \times N^{\text{train}}}$ is the covariance matrix of outputs of all training nodes and $K_{\star D} \in \mathbb{R}^{N^{\text{test}} \times N^{\text{train}}}$ is the covariance between test data and training data. Finally, $\mathbb{I}$ is here the identity in $\mathbb{R}^{N^{\text{train}} \times N^{\text{train}}}$.

## 3.3 Feature distance

To measure and quantify how much a given GCN instance oversmoothes we use the squared Euclidean distance between pairs of nodes, and normalize by the number of node features $d_l$ so that the measure

stays finite in the GP limit $d_l \to \infty$. This allows us to quantitatively test the predictions of our approach on the node-resolved distances of features. To summarize the amount of oversmoothing across the GCN, we also define the measure $\mu(X)$ as the average squared Euclidean distance across all pairs of nodes:

$$d(\boldsymbol{x}_\alpha, \boldsymbol{x}_\beta) = \frac{1}{d_l} ||\boldsymbol{x}_\alpha - \boldsymbol{x}_\beta||_2^2 = C'_{\alpha\alpha} + C'_{\beta\beta} - 2C'_{\alpha\beta}, \tag{13}$$

$$\mu(\boldsymbol{X}) = \frac{1}{2N(N-1)} \sum_{\alpha=1}^{N} \sum_{\beta=\alpha+1}^{N} d(\boldsymbol{x}_\alpha, \boldsymbol{x}_\beta). \tag{14}$$

Here $C'_{\alpha\beta} = \frac{\boldsymbol{x}_\alpha \cdot \boldsymbol{x}_\beta}{d_l}$ is the normalized scalar product. We use the notation $C'_{\alpha\beta}$ to avoid confusion with the expectation value $C_{\alpha\beta}$ defined in the GCN GP (8). In the infinite feature dimensions limit, the quantities $C'_{\alpha\beta}$ in Equation (14) converge to the GCN GP quantities $C_{\alpha\beta}$ defined by (8). In the following sections we will therefore use the $C_{\alpha\beta}$ as predictions for the $C'_{\alpha\beta}$ of finite-size GCNs. The normalization for $d(\boldsymbol{x}_\alpha, \boldsymbol{x}_\beta)$ and $\mu(\boldsymbol{X})$ can be interpreted as an average (squared) feature distance, independent of the size of the graph and the number of feature dimensions.

## 4 Results

### 4.1 Propagation depths

We are interested in analyzing GCNs at large depth. We a priori assume that at infinite depth the GCN converges to an equilibrium in which covariances are static over layers $K_{\alpha\beta}^{(l)} \xrightarrow{l\to\infty} K_{\alpha\beta}^{\text{eq}}$, irrespective of whether the GCN is in the oversmoothing or the chaotic phase. A posteriori we show that this assumption indeed holds. Since the fixed point $K^{\text{eq}}$ is independent of the input, a GCN at equilibrium cannot use information from the input to make predictions (although, as we will see, in the non-oversmoothing phase it can still use the graph structure). In the following we analyze the equilibrium covariance $K^{\text{eq}}$ to which GCNs with different $\sigma_w^2$, $\sigma_b^2$ and $A$ converge to, how they behave near this equilibrium, and at which rate it is approached.

Close to equilibrium, the covariance matrix $K^{(l)}$ can be written as a perturbation around $K_{\alpha\beta}^{\text{eq}}$:

$$K_{\alpha\beta}^{(l)} = K_{\alpha\beta}^{\text{eq}} + \Delta_{\alpha\beta}^{(l)}. \tag{15}$$

Under the assumption that the perturbation $\Delta_{\alpha\beta}^{(l)}$ is small, we can linearize the GCN GP

$$\Delta_{\alpha\beta}^{(l+1)} = \sum_{\gamma,\delta} H_{\alpha\beta,\gamma\delta} \Delta_{\gamma\delta}^{(l)} + \mathcal{O}((\Delta^{(l)})^2), \tag{16}$$

$$H_{\alpha\beta,\gamma\delta} = \sigma_w^2 \sum_{\theta,\phi} \frac{1}{2}(1 + \delta_{\gamma,\delta}) A_{\alpha\theta} A_{\beta\phi} \frac{\partial C_{\theta\phi}}{\partial K_{\gamma\delta}} [K^{\text{eq}}], \tag{17}$$

where we use square brackets to denote the point around which we linearize. The factor $\frac{1}{2}(1 + \delta_{\gamma,\delta})$ is introduced to correctly count on- and off-diagonal elements of the covariance matrix, while the shift operators $A$ and the derivative $\frac{\partial C_{\theta\phi}}{\partial K_{\gamma\delta}}[K^{\text{eq}}]$ originate from the message passing and the non-linearity $\phi$, respectively. The latter would result in a Kronecker delta $\frac{\partial C_{\theta\phi}}{\partial K_{\gamma\delta}}[K^{\text{eq}}] = \delta_{\theta\phi,\gamma\delta}$ for linear networks. The calculation for $H_{\alpha\beta,\gamma\delta}$ is done in detail in Appendix B.

A conceptually similar linearization has been done in [37] for DNNs. In the DNN case, different inputs to the networks—which correspond to input features on different nodes here—can be treated separately, leading to decoupling of Equation (16). The shift operator in the GCN dynamics, in contrast, couples features on neighboring nodes – the matrix $H_{\alpha\beta,\gamma\delta}$ is in general not diagonal.

We can still achieve a decoupling by interpreting Equation (16) as a matrix multiplication, if $\alpha\beta$ and $\gamma\delta$ are understood as double indices, and by finding the eigendirections of the matrix $H \in \mathbb{R}^{N^2 \times N^2}$. Taking the right eigenvectors $V_{\alpha\beta}^{(i)}$ as basis vectors, we can decompose the covariance matrix $\Delta_{\alpha\beta}^{(l)} = \sum_i \Delta_i^{(l)} V_{\alpha\beta}^{(i)}$ and thus obtain the overlaps $\Delta_i^{(l)}$ which evolve independently over layers. If the fixed

point $K^{\mathrm{eq}}$ is attractive, all eigenvalues have absolute values smaller than one: $|\lambda_i| < 1$. This allows us to define the propagation depth $\xi_i := -\frac{1}{\ln(\lambda_i)}$ for each eigendirection, very similar to the DNN case [37]. In this form, the linear update equation (16) simplifies to

$$\Delta_i^{(l+d)} = \lambda_i^d \Delta_i^{(l)} = \exp(-d/\xi_i)\Delta_i^{(l)} \,, \tag{18}$$

thus decoupling the system. For details on the linearization and some properties of the transition matrix $H$ refer to Appendix B.

## 4.2 The non-oversmoothing phase of GCNs

In this section we establish the chaotic, non-oversmoothing phase of GCNs, and show that this phase can be reached by simple tuning of the weight variance $\sigma_w^2$ at initialization. We start by noticing that a GCN is at a state of zero feature distance $\mu(\boldsymbol{X}^{(l)}) = 0$, if the covariance matrix has constant entries, $K_{\alpha\beta}^{(l)} = k^{(l)}$: Constant entries in $K_{\alpha\beta}^{(l)}$ imply that all preactivations are the same, $\boldsymbol{h}_\alpha^{(l)} = \boldsymbol{h}_\beta^{(l)}$, which in turn implies $C_{\alpha\beta}^{(l)} = c^{(l)}$ (by Equation (8)); the latter is equivalent to features being the same, $\boldsymbol{x}_\alpha^{(l)} = \boldsymbol{x}_\beta^{(l)}$. Due to our choice of the shift operator, the state of zero distance (and thus of $K_{\alpha\beta}^{(l)} = k^{(l)}$) is always a fixed point. Assuming that $C_{\alpha\beta}^{(l)} = c^{(l)}$, we obtain

$$K_{\alpha\beta}^{(l+1)} = \sigma_b^2 + \sigma_w^2 \underbrace{\sum_\gamma A_{\alpha\gamma}}_{=1} \underbrace{\sum_\delta A_{\beta\delta}}_{=1} c^{(l)} = k^{(l+1)} \,. \tag{19}$$

In an overmoothing GCN, this fixed point is also attractive, meaning that also pairs of feature inputs $\boldsymbol{x}_\alpha^{(0)}$, $\boldsymbol{x}_\beta^{(0)}$ which initially have non-zero distance $d(\boldsymbol{x}_\alpha^{(0)}, \boldsymbol{x}_\beta^{(0)}) \neq 0$ (and thus $\mu(\boldsymbol{X}^{(0)}) \neq 0$) eventually converge to the point of vanishing distance. The chaotic, non-oversmoothing phase of a GCN is determined by the condition that this point of constant covariance $K_{\alpha\beta}^{(l)} = k^{(l)}$ becomes unstable. More formally, this can be written in terms of eigenvalues of the linearized dynamics as

$$\max\{|\lambda_i^{\mathrm{p}}|\} \overset{?}{>} 1 \,. \tag{20}$$

Here and in the following we will use the superscript p to denote that the linearization is done around the state of constant covariance across nodes in both the oversmoothing and non-oversmoothing phase. The propagation depth $\xi_i := -\frac{1}{\ln(\lambda_i)}$ diverges at the phase transition where one $\lambda_i$ approaches 1. Intuitively speaking, Equation (20) asks whether a small perturbation from the zero distance case diminishes ($\max\{|\lambda_i^{\mathrm{p}}|\} < 1$), in which case the network dynamics is regular, or grows ($\max\{|\lambda_i^{\mathrm{p}}|\} > 1$), in which case the network is chaotic and thus does not oversmooth. The value of $\max\{|\lambda_i^{\mathrm{p}}|\}$ depends on the choices of $A$, $\sigma_w^2$ and $\sigma_b^2$ (by the dependence of $K^{\mathrm{eq}}$ on $\sigma_b^2$). In the following we will concentrate on tuning $\sigma_w^2$ to reach the non-oversmoothing phase.

### 4.2.1 Complete graph

To illustrate the implications of the analysis described above, we first consider a particularly simple GCN on a complete graph; this allows us to calculate the condition for the transition to chaos analytically, and gain some insight into the interesting parameter regimes. Moreover, we use this pedagogical example to show that although the GP equivalence is only true in the limit of infinite hidden feature dimensions, $d_l \to \infty$, our results still describe finite-size GCNs well.

For a complete graph with adjacency matrix $\mathcal{A}_{\alpha\beta} = 1 - \delta_{\alpha\beta}$, our choice of shift operator $A$ in (6) has entries $A_{\alpha\beta} = \frac{g}{N-1} + \delta_{\alpha\beta}(1 - \frac{Ng}{N-1})$. This model is a worst-case scenario for oversmoothing, since the adjacency matrix leads to inputs that are shared across all nodes of the network. We make the ansatz that the equilibrium covariance is of the form $K_{\alpha\beta}^{\mathrm{eq}} = K_c^{\mathrm{eq}} + \delta_{\alpha\beta}(K_a^{\mathrm{eq}} - K_c^{\mathrm{eq}})$ due to symmetry which reduces the problem to only two variables. In this formulation we can use similar methods as in the DNN case [37] to determine the non-oversmoothing condition on the l.h.s in (20) (Details are given in Appendix C).

Figure 1 shows how a GCN on a complete graph can be engineered to be non-oversmoothing by simple tuning of the weight variance $\sigma_w^2$. Panel a) shows that increasing the weight variance $\sigma_w^2$ or decreasing

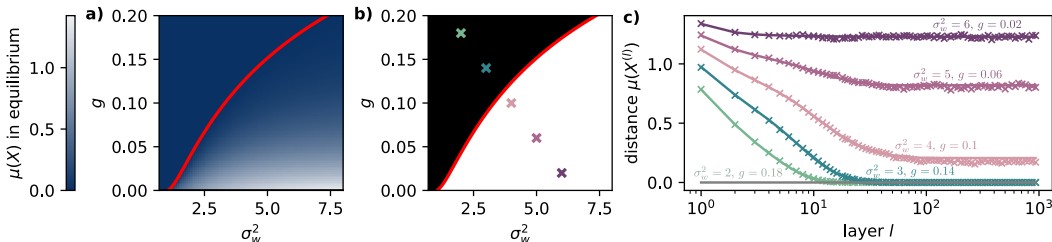

Figure 1: Simulations and GP prior of a GCN on a complete graph with $N = 5$ nodes, shift operator $A_{\alpha\beta} = \frac{g}{N-1} + \delta_{\alpha\beta}(1 - \frac{Ng}{N-1})$, vanishing bias $\sigma_b^2 = 0$ and $\phi(x) = \mathrm{erf}(\frac{\sqrt{\pi}}{2}x)$. **a)** The phase diagram dependent on $\sigma_w^2$ and $g$. The equilibrium feature distance $\mu(\boldsymbol{X})$ obtained from computing the GCN GP prior for $L = 4,000$ layers is shown as a heatmap, the red line is the theoretical prediction for the transition to the non-oversmoothing phase. **b)** Same as in a) but color coding shows whether $\mu(\boldsymbol{X})$ is close to zero (black) or not (white) with precision $10^{-5}$. The red line again shows the theoretically predicted phase transition. **c)** Feature distance $\mu(\boldsymbol{X}^{(l)})$ for a random input $X_{\alpha i}^{(0)} \overset{\text{i.i.d.}}{\sim} \mathcal{N}(0, 1)$ as a function of layer $l$. Parameters are written in the panel in matching colors and marked with color coded crosses in the phase diagram in panel b). Feature dimension of the hidden layers is $d_l = 200$, crosses show the mean of 50 network realizations, solid curves the theoretical predictions.

the size of the off-diagonal elements $g$ both shift the network towards the non-oversmoothing phase. Both parameters also increase the equilibrium feature distance beyond the transition. The theoretical prediction for the transition is calculated in Appendix C and shown as the red line. Panel b) confirms the accuracy of this calculation. Larger values of $g$ increase smoothing, and thus larger values of $\sigma_w^2$ are needed to compensate. Moreover, our formalism allows us to predict the evolution of feature distances over layers correctly, as can be confirmed in panel c). We find again that GCNs with parameters past the transition do not oversmooth.

### 4.2.2 General graphs

For general graphs, the transition to the non-oversmoothing phase given by Equation (20) can be determined numerically. As a proof of concept, we demonstrate this approach for the Contextual Stochastic Block Model (CSBM) [8], a common synthetic model which allows generating a graph with two communities and community-wise correlated features on the nodes. Pairs of nodes within the same community have higher probability of being connected and have feature vectors which are more strongly correlated, compared with pairs of nodes from different communities.

Given the underlying graph structure, we can construct the linear map $H$ from Equation (16) and the analytical solution for $C_{\theta\phi}$ in Appendix A. Finding the set of eigenvalues is then a standard task. We show the applicability of our formalism in Figure 2 by showing that GCNs degenerate to a zero distance state state exactly when $\sigma_w^2 < \sigma_{w,\mathrm{crit}}^2$. Panel a) shows how this procedure correctly predicts the transition in the given CSBM instance: The maximum feature distance between any pair of nodes increases from zero at the point where the state $K_{\alpha\beta}^{(l)} = k^{(l)}$ becomes unstable. This means that beyond this point, the GCN has feature vectors that differ across nodes and therefore does not oversmooth. This is more explicitly shown in panels b) and c), where the equilibrium feature distance is plotted as a heatmap. At point $A$ (panel b)), within the oversmoothing phase, all equilibrium feature distances are indeed zero, the network therefore converges to a state in which all features are the same. At point $B$ (panel c)) on the other hand, pairs of nodes exist that have finite distance. In the latter case, one can recognize the community structure of the CSBM: the lower left and upper right quadrants are lighter than the diagonal ones, indicating larger feature distances across communities than within. The equilibrium state thus contains information about the graph topology. This phenomenon is also observed in panel d), where we show the predicted feature distance averaged for nodes within or between classes as a function of layers compared to finite-size simulations. Again, theoretical predictions match with simulations. Thus also on more general graphs the presented formalism predicts the transition point between the oversmoothing and the non-oversmoothing phase, corresponding to a transition between regular and chaotic behavior.

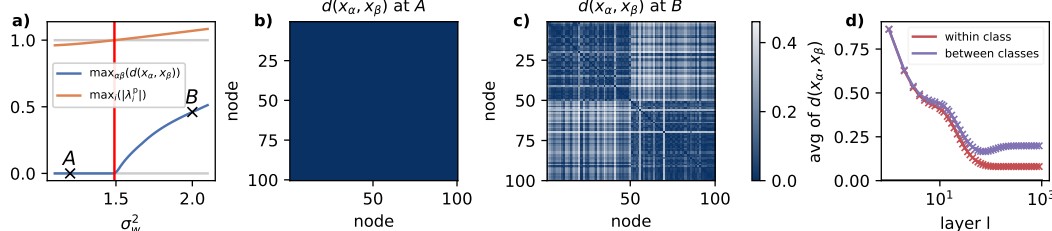

Figure 2: The non-oversmoothing phase in a contextual stochastic block model instance with parameters $N = 100$, $d = 5$, $\lambda = 1$. The shift operator is chosen according to (6) with $g = 0.3$, and $\sigma_b^2 = 0$ and $\phi(x) = \text{erf}(\frac{\sqrt{\pi}}{2}x)$. **a)** The maximum feature distance between any pair of nodes in equilibrium obtained from computing the GCN GP prior for $L = 4,000$ layers (blue) and the largest eigenvalue of the linearized GCN GP dynamics at the zero distance state as a function of weight variance $\sigma_w^2$. The red line marks the point where $\max_i\{|\lambda_i^P|\} = 1$. **b)** Heatmap of the equilibrium distance matrix with entries $d_{\alpha\beta} = d(\boldsymbol{x}_\alpha, \boldsymbol{x}_\beta)$ (Equation (13)) at $\sigma_w^2 = 1.3$, marked as point $A$ in panel a). Colorbar shared with the plot in c). **c)** Same as b) but at point $B$ with $\sigma_w^2 = 2$. **d)** Features distances $d_{\alpha\beta}^{(l)} = d(\boldsymbol{x}_\alpha^{(l)}, \boldsymbol{x}_\beta^{(l)})$ as a function of layers for random inputs $X_{\alpha i}^{(0)} \overset{\text{i.i.d.}}{\sim} \mathcal{N}(0,1)$ and a finite-size GCN with $d_l = 200$, averaged for distances for pairs of nodes within the same community (red) and across communities (purple).

We discuss how the assumptions on weight matrices in related theoretical work [2, 45] exclude networks in the chaotic phase in Appendix D, explaining why the non-oversmoothing phase has not been reported before. In Appendix E we observe how increasing the weight variance increases the oversmoothing measure $\mu(X)$ in equilibrium also in the case of the more common shift operator proposed in the original work [19], despite the fact that this shift operator does not have the oversmoothed fixed point in the sense of Equation (19).

### 4.3 Implications for performance

Lastly we want to investigate the implications of the non-oversmoothing phase on performance. We do this by applying the GCN GP as well as a finite-size GCN to the task of node classification in the CSBM model and measure their performance, shown in Figure 3. Panel a) shows how the generalization error of the GCN GP changes depending on the weight variance $\sigma_w^2$ and the number of layers $L$. In the oversmoothing phase where most GCNs in the literature are initialized (see Appendix D), the generalization error increases significantly already after only a couple of layers. We observe the best performance near the transition to chaos where the GCN GP stays informative up to 100 layers. In panel b) we test the generalization error for even deeper networks. While the generalization error increases to one (being random chance) in the oversmoothing phase, GCN GPs in the chaotic phase stay informative even at more than a thousand layers. This can be explained by Figure 2: For such deep networks, the dynamics are very close to the equilibrium and thus no information of the input features $\boldsymbol{X}^{(0)}$ is transferred to the output. The state, however, still contains information of the network topology from the adjacency matrix, leading to better than random chance performance. In panel c) we explicitly show the layer dependence of the generalization error for the GCN GP at the critical point, in the oversmoothing and in the chaotic phase. Again, we see a fast performance drop for oversmoothing networks, while in the chaotic phase and at the critical point the GCN GP obtains good performance also at large depths, with performance peaking at $L \approx 15$ layers. Tuning the weight variance thus not only prevents oversmoothing, but may also allow the construction of GCNs with more layers and possibly better generalization performance.

In the study of deep networks, results obtained in the limit of infinite feature dimensions $d_l \rightarrow \infty$ often are also applicable for finite-size networks [32, 37]. In panel d) we conduct a preliminary analysis for finite-size GCNs by measuring the performance of randomly initialized GCNs for which we only train the readout layer via gradient descent. Indeed, we observe similar behavior as for the GCN GP: Performance drops rapidly over layers in the oversmoothing phase, while performance stays high over many layers at the critical point and in the chaotic phase and peaks at $L \approx 15$ layers.

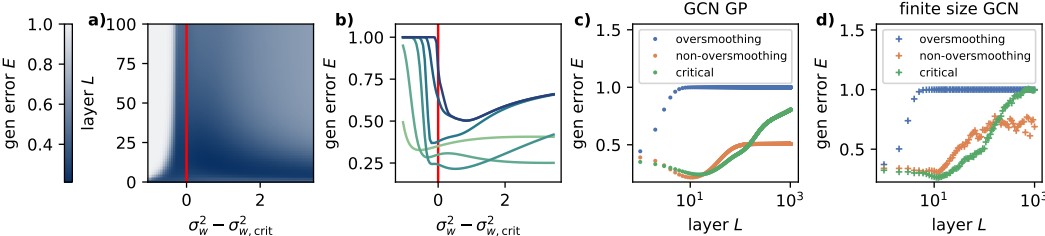

Figure 3: Generalization error (mean squared error) of the Gaussian process for a CSBM with parameters $N = 20$, $d = 5$, $\lambda = 1$, $\gamma = 1$ and $\mu = 4$. The shift operator is defined in (6) with $g = 0.1$, other parameters are $\sigma_b^2 = 0$, $\phi(x) = \mathrm{erf}(\frac{\sqrt{\pi}}{2}x)$ and $\sigma_{ro} = 0.01$. In all panels we use $N^{\mathrm{train}} = 10$ training nodes and $N^{\mathrm{test}} = 10$ test nodes, five training nodes from each of the two communities. Labels are $\pm 1$ for the two communities, respectively. For all panels, we show averages over 50 CSBM instances. **a)** Heatmap of the generalization error of the GCN GP dependent on number of layers $L$ and weight variance $\sigma_w^2$. The red line shows the transition to the non-oversmoothing phase. **b)** Generalization error dependent on weight variance $\sigma_w^2$ and depths $L = 1, 4, 16, 64, 256, 1024$ from turquoise to dark blue. **c)** Generalization error dependent on the layer for the GCN GP at the critical line $\sigma_w^2 = \sigma_{w,\mathrm{crit}}^2$, in the oversmoothing phase $\sigma_w^2 = \sigma_{w,\mathrm{crit}}^2 - 1$ and the non-oversmoothing phase $\sigma_w^2 = \sigma_{w,\mathrm{crit}}^2 + 1$. **d)** Performance of randomly initialized finite-size GCNs with $d_l = 200$ for $l = 1, \ldots, L$ where only the linear readout layer is trained with gradient descent (details in Appendix F) at the critical line $\sigma_w^2 = \sigma_{w,\mathrm{crit}}^2$, in the oversmoothing phase $\sigma_w^2 = \sigma_{w,\mathrm{crit}}^2 - 1$ and the non-oversmoothing phase $\sigma_w^2 = \sigma_{w,\mathrm{crit}}^2 + 1$.

Additionally, we test the performance of the GCN GP on the real world citation network Cora [38]. Evaluating the eigenvalue condition (20) would be computationally expensive for such a large dataset, therefore we find the transition by numerically evaluating the feature distance $\mu(X)$ in equilibrium and search for the $\sigma_w^2$ at which this distance becomes non-zero. This procedure results in $\sigma_{w,\mathrm{crit}}^2 \approx 1$ and is presented in more detail in Appendix G. Figure 4 panel a) shows the performance of the GCN GP dependent on the number of layers $L$ and weight variance $\sigma_w^2$: as for the CSBM in Figure 3 we observe that the performance for deep GCN GPs is best near the transition in the non-oversmoothing phase. Furthermore, GCN GPs with more layers achieve lower generalization error. This is shown more directly in panel b). There we observe the layer dependence for GCN GPs near the transition in the non-oversmoothing regime. Indeed, the accuracy increases up to a hundred layers, reaching the accuracy of finite-size GCNs stated in [19].

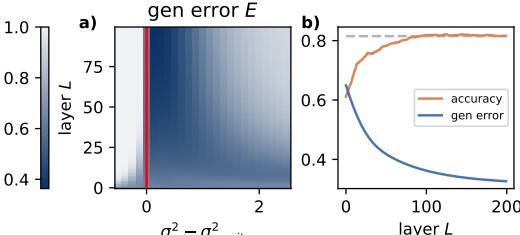

Figure 4: GCN GP performance on the Cora datset [38]. **a)** Generalization error (mean squared error) as a function of layers $L$ and weight variance $\sigma_w^2 - \sigma_{w,\mathrm{crit.}}^2$ for our stochastic shift operator (6) with $g = 0.9$. The value of $\sigma_{w,\mathrm{crit.}}^2 \approx 1$ is determined numerically in Appendix G. **b)** Layer dependent generalization error and accuracy for GCNs near the transition $\sigma_w^2 = \sigma_{w,\mathrm{crit.}}^2 + 0.1$. Grey dashed line shows accuracy obtained for GCNs in the original work [19]. Numerical details in Appendix F.

Near the transition, accuracy increases for up to $L = 100$, and the generalization error improves even beyond this. We hypothesize that this many layers are required for high performance partly due to our choice of the shift operator. The Cora dataset has a maximum degree $d_{\max} = 168$ leading to small off-diagonal elements for the choice of our shift operator: Recall that in Equation (6), the parameter $g$ is constrained to be $g \in (0, 1)$. As a consequence, the off-diagonal elements of the shift operator are $A_{ij} < \frac{1}{d_{\max}} = \frac{1}{168}$. Many convolutional layers are then needed to incorporate information from a node's neighbors.

One might wonder whether it is possible to initialize weights in say the oversmoothing regime, and transition to the non-oversmoothing regime during training. We argue that this is possible in the case of Langevin training (Appendix H).

# 5 Discussion

In this study we used the equivalence of GCNs and GPs to investigate oversmoothing, the property that features at different nodes converge over layers to the same feature vector in an exponential manner. By extending concepts such as the propagation depth and chaos from the study of conventional deep feedforward neural networks [37], we are able to derive a condition to avoid oversmoothing. This condition is phrased in terms of an eigenvalue problem of the linearized GCN GP dynamics around the state where all features are the same: This state is stable if all eigenvalues are smaller than one, thus the networks do oversmooth. If one eigenvalue, however, is larger than one, the state where the features are the same on all nodes becomes unstable. While most GCNs studied in the literature are in the oversmoothing phase [2, 45], the non-oversmoothing phase can be reached by a simple tuning of the weight variance at initialization. An analogy can be drawn between the chaotic phase of DNNs and the non-oversmoothing phase of GCNs. Previous theoretical works have proven that oversmoothing is inevitable in some GNN architectures, among them GCNs; these works, however, make crucial assumptions on the weight matrices, constraining their variances to be in what we identify as the oversmoothing phase. Near the transition, we find GCNs which are both deep and expressive, matching the originally reported GCN performance [19] on the Cora dataset with GCN GPs beyond 100 layers.

**Limitations.** The current analysis is based on the equivalence of GCNs and GPs which strictly holds only in the limit of infinite feature dimension. GCNs with large feature vectors ($d_l = 200$) are well described by the theory, as shown in Section 4. For a small number of feature dimensions, however, we expect deviations from the asymptotic results. Throughout the main part of this work, we assumed a row-stochastic shift operator which made the equilibrium $K^{\mathrm{eq}}$ in the oversmoothing phase particularly simple. For other shift operators, we expect qualitatively similar results while the equilibrium $K^{\mathrm{eq}}$ may look different in detail. In our preliminary experiments on the common shift operator from [19] (Appendix E), we indeed find that increasing the weight variance increases the distances between features also in this case. We hypothesize that this effect makes the equilibrium more informative of the graph topology, as in the stochastic shift operator case. The choice of non-linearity is unrestricted, but in the general case numerical integration of (8) is needed.

To determine whether a given weight variance is in the non-oversmoothing phase, one calculates the eigenvalues of the linearized GCN GP dynamics which take the form of an $N^2 \times N^2$ matrix (see Equation (16)), this has a run time of $\mathcal{O}(N^6)$. While this becomes computationally expensive for large graphs, the conceptual insights of the presented analysis remain. In practical applications with large graphs one may reduce the computational load by determining the transition point via computation of the GCN GP prior until it is close to equilibrium. This procedure has a runtime of $\mathcal{O}(N^3 L^{\mathrm{eq}})$ where $L^{\mathrm{eq}}$ is the number of layers after which the process is sufficiently close to equilibrium. One might then do an interval search on the weight variance until the transition point is determined with sufficient accuracy.

**Oulook.** Formulating GCNs with the help of GPs can be considered the leading order in the number of feature space dimension $d_l$ when approximating finite-size GCNs. Computing corrections for finite numbers of hidden feature dimensions would allow the characterization of feature learning in such networks, similar as in standard deep networks [26, 48, 39]. Moreover, the generalization of this formalism to more general shift operators and other GNN architectures [49, 4] like GATs [41] are possible directions of future research. In the special case of GATs we expect similar results to the GCN analyzed here, since the shift operator is constructed using a softmax and therefore also is row-stochastic.

### Acknowledgements

Funded by the European Union (ERC, HIGH-HOPeS, 101039827). Views and opinions expressed are however those of the author(s) only and do not necessarily reflect those of the European Union or the European Research Council Executive Agency. Neither the European Union nor the granting authority can be held responsible for them. We also acknowledge funding by the German Research Council (DFG) within the Collaborative Research Center "Sparsity and Singular Structures" (SfB 1481; Project A07).

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

## A  Analytical solution for expectation values

To evaluate our theory, we need to compute expectation values given in the form of (8) which we restate here for readability

$$C_{\gamma\delta}^{(l)} = \left\langle \phi\left(h_\gamma^{(l)}\right)\phi\left(h_\delta^{(l)}\right)\right\rangle_{h_\gamma^{(l)},h_\delta^{(l)}}, \tag{21}$$

where $h_\gamma^{(l)}$ and $h_\delta^{(l)}$ are zero mean random Gaussian variables with $\langle h_\gamma^{(l)} h_\delta^{(l)}\rangle = K_{\gamma\delta}^{(l)}$. This can be evaluated numerically for general non-linearities $\phi(x)$. For simplicity, however, we choose $\phi(x) = \mathrm{erf}(\frac{\sqrt{\pi}}{2}x)$ in our experiments where the scaling factor in the erf is chosen such that $\frac{\partial\phi}{\partial x}(0) = 1$. In this case, the expectation value can be evaluated analytically to be

$$C_{\gamma\delta}^{(l)} = \frac{2}{\pi}\ \arcsin\left(\frac{\frac{\pi}{2}K_{\gamma\delta}^{(l)}}{\sqrt{1+\frac{\pi}{2}K_{\gamma\gamma}^{(l)}}\sqrt{1+\frac{\pi}{2}K_{\delta\delta}^{(l)}}}\right). \tag{22}$$

as shown in [44].

## B  The linearized GP of GCNs

We start from the full GCN GP iterative map (8,(9)), which we restate here for readability

$$K_{\alpha\beta}^{(l+1)} = T_{\alpha\beta}[K^{(l)}] = \sigma_b^2 + \sigma_w^2\sum_{\gamma,\delta}A_{\alpha\gamma}A_{\beta\delta}C_{\gamma\delta}^{(l)}[K^{(l)}] \tag{23}$$

where $h_\gamma^{(l)}$ and $h_\delta^{(l)}$ are drawn from a 0-mean Gaussian distribution with covariance $\langle h_\gamma^{(l)} h_\delta^{(l)}\rangle = K_{\gamma\delta}^{(l)}$. The full covariance matrix of layer $l$ is denoted as $K^{(l)}$. Here, we distinguish between the iterative maps $T_{\alpha\beta} : \mathbb{R}^{N\times N} \to \mathbb{R}$ of which there are $N^2$, one for each pair of nodes $\alpha$ and $\beta$, and the entries $K_{\alpha\beta}^{(l+1)}$ of the covariance matrix in the layer $l+1$. Notice that $T_{\alpha\beta} = T_{\beta\alpha}$ due to the symmetry of the covariance matrix. In the maps $T_{\alpha\beta}$, the covariance matrix of the previous layer only shows up in the expectation value $C_{\gamma\delta} = \left\langle \phi\left(h_\gamma^{(l)}\right)\phi\left(h_\delta^{(l)}\right)\right\rangle_{h_\gamma^{(l)},h_\delta^{(l)}}$ such that the linearized dynamics around a fixed point (being equilibrium $K_{\alpha\beta}^{\mathrm{fix}} = K_{\alpha\beta}^{\mathrm{eq}}$ or zero distance state $K_{\alpha\beta}^{\mathrm{fix}} = k$) with $K_{\alpha\beta}^{(l)} = K_{\alpha\beta}^{\mathrm{fix}} + \Delta_{\alpha\beta}^{(l)}$ read

$$K_{\alpha\beta}^{(l+1)} = K_{\alpha\beta}^{\mathrm{fix}} + \Delta_{\alpha\beta}^{(l+1)} = \underbrace{T_{\alpha\beta}[K^{\mathrm{fix}}]}_{=K_{\alpha\beta}^{\mathrm{fix}}} + \sum_{\gamma<\delta}\frac{\partial T_{\alpha\beta}}{\partial K_{\gamma\delta}}[K^{\mathrm{fix}}]\Delta_{\gamma\delta}^{(l)} + \mathcal{O}((\Delta^{(l)})^2) \tag{24}$$

$$= K_{\alpha\beta}^{\mathrm{fix}} + \sum_{\gamma,\delta}\sigma_w^2\underbrace{\sum_{\theta,\phi}\frac{1}{2}(1+\delta_{\gamma,\delta})A_{\alpha\theta}A_{\beta\phi}\frac{\partial C_{\theta\phi}}{\partial K_{\gamma\delta}}[K^{\mathrm{fix}}]}_{\equiv H_{\alpha\beta,\gamma\delta}}\Delta_{\gamma\delta}^{(l)} + \mathcal{O}((\Delta^{(l)})^2), \tag{25}$$

where we restrict the sum in (24) to $\gamma < \delta$ since $K_{\gamma\delta}$ and $K_{\delta\gamma}$ are the same quantity. From (24) follows

$$\Delta_{\alpha\beta}^{(l+1)} = \sum_{\gamma,\delta}H_{\alpha\beta,\gamma\delta}\Delta_{\gamma\delta}^{(l)} + \mathcal{O}((\Delta^{(l)})^2) \tag{26}$$

which is Equation (16) in the main text. While $H$ is not symmetric in general, $H_{\alpha\beta,\gamma\delta} \neq H_{\gamma\delta,\alpha\beta}$, it is symmetric in the first and second pair of covariance indices, $H_{\alpha\beta,\gamma\delta} = H_{\beta\alpha,\gamma\delta}$ and $H_{\alpha\beta,\gamma\delta} = H_{\alpha\beta,\delta\gamma}$ due to symmetry of the covariance matrices, $K_{\alpha\beta} = K_{\beta\alpha}$.

In the main text, we look for the right eigenvectors of $H$ fulfilling

$$\lambda_i V_{\alpha\beta}^{(i)} = \sum_{\gamma,\delta}H_{\alpha\beta,\gamma\delta}V_{\gamma\delta}^{(i)}. \tag{27}$$

These are for general non-symmetric matrices not orthogonal. In order to decompose $\Delta^{(l)}$ to overlaps with the eigenvectors $V_{\alpha\beta}^{(i)}$ we need to find the dual vectors $U_{\alpha\beta}^{(i)}$ fulfilling

$$\sum_{\alpha,\beta} U_{\alpha\beta}^{(i)} V_{\alpha\beta}^{(j)} = \delta_{ij} \tag{28}$$

from which we can define

$$\Delta_i^{(l)} = \sum_{\alpha,\beta} U_{\alpha\beta}^{(i)} \Delta_{\alpha\beta}^{(l)} \tag{29}$$

such that

$$\Delta_{\alpha\beta}^{(l)} = \sum_i \Delta_i^{(l)} V_{\alpha\beta}^{(i)} \tag{30}$$

as stated in the main text.

## C  Investigation of the complete graph model

In this section we analytically investigate the complete graph model as defined in the main text. Specifically, we consider networks with the shift operator

$$A_{\alpha\beta} = \frac{g}{N-1} + \delta_{\alpha\beta}(1 - \frac{Ng}{N-1}) \tag{31}$$

and $\phi = \mathrm{erf}(\sqrt{\pi}x/2)$. Due to the symmetry of the system we make the ansatz

$$K_{\alpha\beta}^{\mathrm{eq}} = K_{\mathrm{a}}^{\mathrm{eq}} + \delta_{\alpha\beta}(K_{\mathrm{c}}^{\mathrm{eq}} - K_{\mathrm{a}}^{\mathrm{eq}}), \tag{32}$$

meaning that we assume constant variances $K_{\mathrm{a}}^{\mathrm{eq}}$ across nodes and that all pairs of nodes have the same covariance $K_{\mathrm{c}}^{\mathrm{eq}}$, reducing the system to only two unknown variables. The equilibrium is a fixed point of the iterative map of the GCN GP. With the special choice of shift operator in (31) this becomes

$$K_{\mathrm{a}}^{(l+1)} = \sigma_b^2 + g_{\mathrm{a}} C_{\mathrm{a}}^{(l)} + g_{\mathrm{c}} C_{\mathrm{c}}^{(l)} \tag{33}$$

and

$$K_{\mathrm{c}}^{(l)} = \sigma_b^2 + h_{\mathrm{a}} C_{\mathrm{a}}^{(l)} + h\prime_{\mathrm{c}} C_{\mathrm{c}}^{(l)} \tag{34}$$

with constants

$$g_{\mathrm{a}} = (1 + \frac{g^2}{N-1})\sigma_w^2 \tag{35}$$

$$g_{\mathrm{c}} = 2(g + \frac{g^2(N-2)}{N-1})\sigma_w^2 \tag{36}$$

$$h_{\mathrm{a}} = 2(\frac{g}{N-1} + \frac{g^2(N-2)}{(N-1)^2})\sigma_w^2 \tag{37}$$

$$h_{\mathrm{c}} = (1 + \frac{g^2}{(N-1)^2} + 2\frac{g^2(N-2)(N-3)}{(N-1)^2} + 4\frac{g(N-2)}{(N-1)})\sigma_w^2 \tag{38}$$

and

$$C_{\mathrm{a}}^{(l)} = \langle \phi(h_\alpha^{(l)}) \phi(h_\alpha^{(l)}) \rangle \tag{39}$$

$$C_{\mathrm{c}}^{(l)} = \langle \phi(h_\alpha^{(l)}) \phi(h_\beta^{(l)}) \rangle \qquad \text{for } \alpha \neq \beta . \tag{40}$$

The preactivations are Gaussian distributed with zero mean and covariance $\langle h_\alpha^{(l)} h_\beta^{(l)} \rangle = K_{\alpha\beta}^{(l)}$. In the oversmoothing phase, we know that $\mu(\boldsymbol{X}) = 0$ in equilibrium. We have seen in Section 4.2 that this corresponds to $K_{\alpha\beta}^{\mathrm{eq}} = k^{\mathrm{eq}}$, implying for our ansatz that $K_{\mathrm{c}}^{\mathrm{eq}} = K_{\mathrm{a}}^{\mathrm{eq}}$. We will find the transition to chaos by calculating where this state becomes unstable with regard to small perturbations. Specifically, we define $c^{(l)} = \frac{K_{\mathrm{c}}^{(l)}}{K_{\mathrm{a}}^{(l)}}$ and look for the parameter point where

$$1 \overset{?}{>} \left. \frac{\partial c^{(l+1)}}{\partial c^{(l)}} \right|_{c^{(l)}=1} . \tag{41}$$

The authors in [37] used this approach to find the transition to chaos for DNNs. The correlation coefficient is

$$c^{(l+1)} = \frac{K_c^{(l+1)}}{K_a^{(l+1)}} = \frac{\sigma_b^2 + h_a C_a^{(l)} + h_c C_c^{(l)}}{\sigma_b^2 + g_a C_a^{(l)} + g_c C_c^{(l)}} \tag{42}$$

and Equation (41) thus becomes

$$\left.\frac{\partial c^{(l+1)}}{\partial c^{(l)}}\right|_{c^{(l)}=1} = \frac{h_c \frac{\partial C_c^{(l+1)}}{\partial c^{(l)}} K_a^{(l+1)} - g_c \frac{\partial C_c^{(l+1)}}{\partial c^{(l)}} K_c^{(l+1)}}{(K_a^{(l+1)})^2}. \tag{43}$$

Since we look at this equation at the perfectly correlated state $c^{(l)} = 1$, we know that $K_a^{(l)} = K_c^{(l)}$ (implying that $C_a^{(l)} = K_c^{(l)}$) and can determine $K_a^{(l)}$ as the solution of the fixed point equation

$$K_a^{(l+1)} = \sigma_b^2 + (g_a + g_c) C_a^{(l)} \tag{44}$$

to which the GCN GP dynamics reduce in the zero distance state (by using the fact that $\sum_\beta A_{\alpha\beta} = 1$ an $C_{\alpha\beta}^{(l)} = c^{(l)}$). Lastly, we can calculate

$$\left.\frac{\partial C_c^{(l+1)}}{\partial c^{(l)}}\right|_{c=1} = \left.\frac{\partial}{\partial c^{(l)}}\left(\frac{2}{\pi}\arcsin\left(\frac{\frac{\pi}{2}K_a^{(l)}c^{(l)}}{1 + \frac{\pi}{2}K_a^{(l)}}\right)\right)\right|_{c^{(l)}=1} \tag{45}$$

$$= \left.\frac{2}{\pi}\frac{1}{\sqrt{1 - \left(\frac{\frac{\pi}{2}K_a^{(l)}c^{(l)}}{1+\frac{\pi}{2}K_a^{(l)}}\right)^2}}\frac{K_a^{(l)}}{\frac{2}{\pi}+K_a^{(l)}}\right|_{c^{(l)}=1} \tag{46}$$

$$= \frac{2}{\pi}\frac{1}{\sqrt{1 - \left(\frac{K_a^{(l)}}{\frac{2}{\pi}+K_a^{(l)}}\right)^2}}\frac{K_a^{(l)}}{\frac{2}{\pi}+K_a^{(l)}}, \tag{47}$$

where we used the known solution for the expectation value $C_c$ for $\phi(x) = \mathrm{erf}(\sqrt{\pi}x/2)$ from Appendix A. Plugging this into Equation (43) lets us calculate $\left.\frac{\partial c^{(l+1)}}{\partial c^{(l)}}\right|_{c^{(l)}=1}$ and thus determine the transition to the non-oversmoothing phase. This is plotted as a red line in Figure 1 panel a) and b).

## D  Restriction of weight matrices in related work

In this section we will show histograms of critical weight variances $\sigma_{w,\mathrm{crit}}^2$ and discuss how the assumptions in [2] and [45] exclude networks in the non-oversmoothing phase. Our results thus stand in no conflict with the results of these works.

Here we want to argue that the assumptions on the weight matrices in related work constrains their architectures to the oversmoothing phase. We start with [45] in which the authors study graph attention networks (GATs). Although we study a standard GCN here, we hypothesize that increasing the weight variance at initialization likewise prevents oversmoothing in other architecture, such as the GAT in [45]. The critical assumption constraining them to the oversmoothing phase is their assumption A3, stating that $\{||\prod_{l=0}^k |W^{(l)}|||_{\max}\}_{k=0}^\infty$ is bounded where $||M||_{\max} = \max_{i,j} |M_{i,j}|$. For our setting of randomly drawn weight matrices $W_{ij}^{(l)} \overset{\mathrm{i.i.d.}}{\sim} \mathcal{N}(0, \frac{\sigma_w^2}{d_l})$ with $W_{ij}^{(l)} \in \mathbb{R}^{N \times N}$, this restricts us to values $\sigma_w^2 \leq 1$. This can be seen by using the circular law from random matrix theory [11]: It is known that the eigenvalues of a matrix with i.i.d. random Gaussian entries of the form above have eigenvalues uniformly distributed in a circle around 0 in the complex plane with radius $\sqrt{\sigma_w^2}$ in the limit $d_l \to \infty$. Thus, the maximal real part of any eigenvalue of this matrix is $\sqrt{\sigma_w^2}$. Thus we can estimate $||\prod_{l=0}^k |W^{(l)}|||_{\max} \leq c(\sqrt{\sigma_w^2})^k$ with a constant $c$. To enter the non-oversmoothing phase, we need $\sigma_w^2 > 1$. In this case, the latter expression diverges for $k \to \infty$, thus being excluded by the proof in [45]. Indeed, for the CSBMs we investigated in this work, all critical weight variances $\sigma_{w,\mathrm{crit}}^2$ are larger than 1 as shown in Figure 5. Also in our model of the complete graph and the CSBM investigated in Section 4.2.2 all $\sigma_{w,\mathrm{crit}}^2$ are larger than 1, compare Figure 1 and Figure 2.

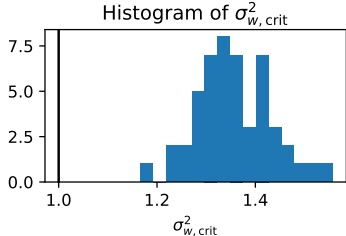

Figure 5: Histogram of $\sigma^2_{w,\mathrm{crit}}$ for the $50$ CSBM instances used in the experiment of Figure 3. The point 1 is marked for comparison to related work.

The authors of [2] and [30] also study GCNs,; however they consider a different shift operator than in this work. In both of [2] and [30] the authors find that their GCN models exponentially loose expressive power if $s\overline{\lambda} < 1$, with $\overline{\lambda}$ being the maximal singular value/eigenvalue of the shift operator and $s$ being the maximal singular value of all weight matrices. Again, the maximal singular value is limited (dependent on $\overline{\lambda}$), which in our approach translates to a limit on $\sigma^2_w$. While the authors notice that their bounds do not hold for large singular values $s$, they do not observe a non-oversmoothing phase in their models.

# E   Non-oversmoothing GCNs with non-stochastic shift operators

Here we investigate how increasing the weight variance can mitigate oversmoothing also in the case of the original shift operator proposed by Kipf and Welling in [19] being

$$A_{\mathrm{KW}} = (D')^{-1/2}\mathcal{A}'(D')^{-1/2} \tag{48}$$

with $\mathcal{A}' = \mathbb{I} + \mathcal{A}$ and $D'_{ij} = \delta_{ij}\sum_k \mathcal{A}'_{ik}$. This shift operator $A_{\mathrm{KW}}$ is not row-stochastic, therefore the state $K_{\alpha\beta} = k$ is not a fixed point for $k \neq 0$, as can be seen from Equation (19). Thus we need to differentiate between two kind of fixed points: Either, all features are zero, for which $K_{\alpha\beta} = 0$ and $\mu(X) = 0$, or some $K_{\alpha\beta} \neq 0$ in which case we have $\mu(X) > 0$. Importantly, for this shift operator, there is no intermediate case for which $K_{\alpha\beta} = k$ with $k \neq 0$. Consequently there is no oversmoothing regime with respect to the measure $\mu(X)$ where node features are non-zero. For an in depth study of this case (48), the definition of another oversmoothing measure $\mu'$ incorporating the different values of $\sum_\beta (A_{\mathrm{KW}})_{\alpha\beta}$ for different $\alpha$ may be more appropriate. For our purposes, it will suffice to analyze the shift operator (48) with the measure $\mu(X)$ from Equation (14).

Figure 6 panel a) shows how $\mu(X)$ in equilibrium increases for larger weight variances $\sigma^2_w$ for the shift operator (48). For comparison, we also show the results obtained with our row-stochastic shift operator (6). Thus we find that also for the shift operator $A_{\mathrm{KW}}$ the pairwise distance between features can be increased by increasing the weight variance $\sigma^2_w$. The non-existence of an oversmoothed fixed point except for the special case $K_{\alpha\beta} = 0$ which we argued for above is observed in panel b) and c): The oversmoothing measure $\mu(X)$ is zero for $A_{\mathrm{KW}}$ if and only if all entries of the covariance matrix are zero, implying that all preactivations and thus also all features are zero. This is a qualitative difference to our shift operator $A$ for which we find equilibrium states with non-zero $\max_\alpha K^{\mathrm{eq}}_{\alpha\alpha}$ but still $\mu(X) = 0$.

# F   Details of numerical experiments

To conduct our experiments we use NumPy [13], SciPy [42] (both available under a BSD-3-Clause License) and Scikit-learn (sklearn) [31] (available under a New BSD License). The code is publicly available under `https://github.com/bepping/non-oversmoothing-gcns`. For our experiments with the Cora dataset, we use the readin methods from [19] which are available under a MIT license (Copyright (c) 2016 Thomas Kipf). Computations were performed on CPUs.Requirements for the experiments with synthetic data are (approximately):

- Figure 1: 10mins on a single core laptop.

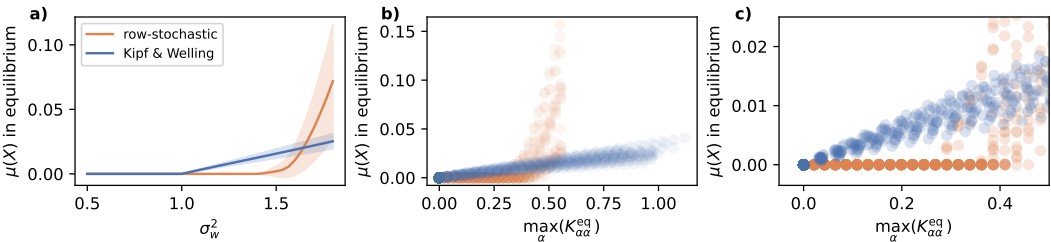

Figure 6: Oversmoothing in GCN GPs with the commonly used shift operator (48) (called Kipf & Welling in the label) and our row-stochastic shift operator (6). **a)** Feature distance $\mu(X)$ in equilibrium dependent on weight variance $\sigma_w^2$ obtained from simulating the GCN GP priors for $L^{\mathrm{eq}} = 4,000$ layers. Shown are the averages (solid lines) and standard deviations (shaded areas) over 20 CSBM initializations with $\lambda = 1$, $d = 5$ and $N = 30$ nodes. **b)** Scatter plot of maximal variance of all nodes $\max_\alpha \{K_{\alpha\alpha}^{\mathrm{eq}}\}$ and feature distance $\mu(X)$ in equilibrium for the same data as in plot a). **c)** Same as b) but zoomed into lower left corner.

- Figure 2: 10h on a single node on an internal CPU cluster. Most of the computation time is needed for evaluating $\max(\lambda_i^{\mathrm{p}})$ in panel a).
- Figure 3: 2h on a single core laptop. In panel d), the last layer of finite-size GCNs is trained with the standard settings from sklearn.linear_model.SGDRegressor().
- Figure 5: Byproduct of Figure 2.
- Figure 6: 10min on a single core laptop.

We also experimented on the real world benchmark dataset Cora [38]. This is a citation network with 2708 nodes, representing publications, and 5429 edges, representing the citations between them (we use undirected edges). The publications are divided into seven classes, and the task is to predict these classes for the unlabeled nodes. Node features are of 0/1 valued vectors indicating the absence/presence of words from a dictionary in the titles of the respective publication. The dictionary consists of 1433 unique words. Requirements for the experiments with the Cora dataset are:

- Figure 4: 1h on a single core laptop.
- Figure 7: 15h on a single core laptop.

## G   Non-oversmoothing transition in the Cora dataset

For Figure 4 we numerically determined the transition to the non-oversmoothing regime for the Cora dataset. The transition was estimated by measuring the distance $\mu(X)$ at equilibrium (i.e., after many layers), and determining at which value of $\sigma_w^2$ it becomes larger than a small distance $\epsilon = 10^{-5}$. The results of this experiment are shown in Figure 7. We find the critical point to be $\sigma_{w,\mathrm{crit}}^2 = 1$ while using a step size of $\delta\sigma_w^2 = 0.01$.

## H   Transitioning between regimes during training

In this section argue why we think it is possible to transition from the oversmoothing to the non-oversmoothing regime or vice versa during training.

In the GP limit of infinitely many hidden features with a finite amount of training data, the variance of weights of a neural network is the same before and after training. This is because weights only change marginally in this limit, also known as the lazy training regime [16, 10]. We will use this fact to argue that Langevin training is capable of transitioning from one regime to the other.

Langevin training is a gradient based training scheme with external noise and a decay term. The gradient flow equation is given as

$$\frac{\mathrm{d}W_{ij}}{\mathrm{d}t} = -\gamma W_{ij} - \nabla_{W_{ij}} L + B_{ij} \,,$$

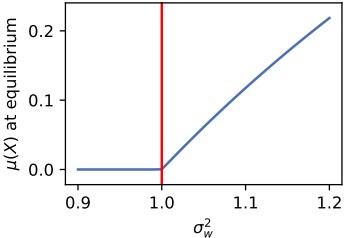

Figure 7: Node distance measure $\mu(\boldsymbol{X})$ at equilibrium obtained from computing the GCN GP prior for $L = 4,000$ layers as a function of $\sigma_w^2$. The transition to the non-oversmoothing regime is estimated by checking where the node distance measure is larger than $\epsilon = 10^{-5}$, marked as the red line.

where $W_{ij}$ are the weights to be trained, $L$ is the loss function and $B_{ij}$ denotes external white noise $\langle B_{ij}(t)B_{kl}(s)\rangle = \delta_{i,k}\delta_{j,l}\delta(t-s)D$ with strength $D$ where $\delta_{a,b}$ and $\delta(a-b)$ denote the Kronecker or Dirac delta, respectively. Both $\gamma$ and $D$ are parameters of this training scheme. It is known that the distribution of weights converges to the posterior weight distribution of a neural network GP with weight variance $\sigma_w^2 = \frac{D}{2\gamma}$ in the infinite feature limit [25] which, as we have noticed above, is the same as the prior distribution.

Since the Langevin distribution converges to the GP weight posterior for any initial distribution, one can imagine an initial distribution with a variance that initializes the network in the oversmoothing regime, while the parameters $\gamma$ and $D$ are chosen such that $\sigma_w^2 = \frac{D}{2\gamma} > \sigma_{w,\text{crit}}^2$ implying that after training most weight realizations will be in the non-oversmoothing regime. Thus, in this case the GCN would have transitioned from one regime to the other. While this argument is made in the limit of infinitely many hidden features, we think that qualitatively similar results are possible for a large but finite number of hidden feature dimensions. The transition in the reverse direction is possible by the same argument only with the initial and the final variances exchanged.

