# OpenReview forum: "Graph Neural Networks Do Not Always Oversmooth"
_NeurIPS.cc/2024/Conference — NeurIPS 2024 poster_

### Official Review · Reviewer_WQpY · 2024-07-11

**Soundness:** 3
**Presentation:** 3
**Contribution:** 3
**Rating:** 6
**Confidence:** 3

**Summary:**

This paper studies the over-smoothing effect in Graph Convolution Networks in the infinite width limit using their Gaussian processes equivalence. The authors generalize the concept of deep information propagation to GCNs and identify the similarity between ordered and chaotic phases in deep neural networks to over-smoothing and non over-smoothing regimes in GCNs. Using this, they show that initializing weights with large variance is sufficient to escape over-smoothing in GCNs.

**Strengths:**

1. The theory of using deep information propagation to graphs is novel and interesting. Although many works studied over-smoothing in GNNs, this paper analyzes it from a new perspective resulting in interesting findings.
2. The paper is well written and structured. I appreciate the authors discussed the implications and limitations of their work in detail.

**Weaknesses:**

1. The analysis is done using column shift stochastic shift operator, but in practice, GCNs are usually constructed with degree normalized adjacencies such as $D^{-1/2}AD^{-1/2}$ or $D^{-1}A$. I wonder if the results would translate in this case as well. The authors have discussed this in the limitations, perhaps giving empirical evidence using CSBM could help counter this weakness.
2. The theoretical analysis is in infinite width limit so it may not hold well in the finite width setting. But in my opinion this is not a critical weakness of the paper as the analysis in finite width is more challenging.

**Questions:**

Please refer to the Weaknesses for major comments and minor ones are listed below.

1. To strengthen the paper further, I suggest the authors evaluate their findings on at least one real graph data like Cora or Citeseer. I understand the computational difficulties, and to tackle it, one can consider a small labeled set.
2. Would the result extend to other graph neural networks like GAT too?
2. What is the significance of considering $\epsilon$ noise in $Y$ (eq. 2)?

**Limitations:**

Limitations are already discussed in the paper which I list below. I don't see any other additional limitations.

1. The analysis is in infinite width limit.
2. The considered shift operator is different from the one usually used in practice.
3. Computational disadvantages in determining the variance of weights.

---

> ### Author Rebuttal · Authors · 2024-08-07
>
> We are grateful for the reviewer’s insightful comments and the evaluation of our
> work. The comments are very helpful; we will address them in the revision:
>
> Re: Weaknesses
> 1. This is a good suggestion; indeed it is something we are currently looking
> into. These shift operators are not column-stochastic, so from the point
> of view of the theory they present additional challenges. However there
> is no issue to repeating our numerical experiments with these operators,
> so we will perform the suggested numerical experiments, and add them
> to the appendix. If we find compelling theoretical results for these more
> general operators we may add them to the main text.
> 2. Indeed the extension to finite networks is challenging and outside of our
> scope. In the discussion we point to some works which perform finite-size
> corrections for deep neural networks, and could serve as a starting point
> for a similar extension to GCNs. Nevertheless, already the infinite width
> limit may give good qualitative insights; this is certainly the case for wide
> deep neural networks, and based on our results seems also to be the case
> for GCNs (Compare our Figs 3c and 3d). For the wide networks we consider here, the Gaussian process is an accurate description of the posterior
> after training (Compare our Figs 1c, 2d, and 3c and 3d, .).
> The more interesting setting, which we guess is the one referred to by
> the referee, is that of networks of small or moderate width. While the
> weights in wide networks change during training only marginally, in small
> networks trained weights typically depart from their values at initialization. Still in this regime we expect the found criterion for the start of the
> non-oversmoothing regime to be indicative of the network’s smoothing
> properties.
>
> Re: Questions
> 1. In the revision we will repeat the experiments shown in Fig. 3 with a
> shift operator determined from the Cora and/or CiteSeer datasets. For
> preliminary results on the Cora dataset, please refer to our global response.
> 2. We expect that our results generalize to other architectures like graph attention networks (GATs); in fact the shift operator in GATs is defined in
> terms of a softmax (cf. [1] Eq. (2)), and therefore satisfies our assumption that it be column-stochastic. However it also depends on learnable
> parameters, which would require a generalization of our formalism.
> More broadly the question of generalizing our theory to alternative architectures is a valid and interesting one, and we plan to expand the relevant
> section in the discussion to include both GATs and operators based on the
> normalized Laplacian.
> 3. The readout noise is kept for numerical reasons: In the oversmoothing
> phase, the entries of the covariance matrix converge to constant values $K_{\alpha\beta}^{(l)} \overset{l\to\infty}{\to} k$. To infer labels of test nodes however, we need to invert the
> covariance matrix $K_{\alpha\beta}$ in Eq. (12). Adding readout noise amounts to
> adding a small diagonal matrix, which stabilizes the numerics. We will
> add a comment explaining this in the revision.
>
> References:
>
> [1] Veličković, Petar, et al. "Graph attention networks." arXiv preprint
> arXiv:1710.10903 (2017).

---

> > ### Comment · Reviewer_WQpY · 2024-08-09
> >
> > I thank the authors for the clarifications, and as there were no major concerns in my initial review, I retain my score.

---

### Official Review · Reviewer_kVv2 · 2024-07-15

**Soundness:** 4
**Presentation:** 4
**Contribution:** 3
**Rating:** 7
**Confidence:** 3

**Summary:**

The paper investigates whether oversmoothing in GNNs is avoidable in theory. It investigates why oversmoothing occurs and derives an affirmative answer. In particular, depending on the variance of the initial weight matrices, GNNs can either enter a "regular" (oversmoothing) or "chaotic" phase (non-oversmoothing). These theoretical findings are confirmed by experiments on small-scale synthetic graphs.

**Strengths:**

- The work focuses on a fundamental problem in GNNs, which is introduced and motivated well.
- The background is concise yet easy to follow.
- The paper questions the establishing belief that oversmoothing is inevitable, making this work highly significant.
- The experiments are insightful and presented well.
- The approach is original, and the presentation is clearly written and high-quality.

**Weaknesses:**

- The transfer of the newly developed theory to practical networks is minimal. Sec. 4.3 still considers a CSBM instead of real-world data.
- The work only considers GCNs.
- The derivation of Eq. (17) is a bit hard to follow without further explanation.
- No implementation is provided. No reason is given for why that is the case (cf. l. 529). However, the code for the experiments is also not the work's main contribution.

Minor comments:
- The reason for the specific definition of $\phi$ in l. 120 is only given in l. 148. This could be referenced earlier.
- I assume that the sum in (16) is over all $\gamma$ and $\delta$. If so, a comma would help clarify that. Analogous in (17).
- The order of the appendices A and B is reversed with respect to the presentation in the main paper.
- The captions of the experiment figures are somewhat redundant from the main text.
- One could also directly label the lines in Fig. 1c (instead of in the caption), making it more easily readable. Similar for Fig. 3b.

**Questions:**

1. Regarding l. 175: Is the fixed point intentionally equal for all $\alpha,\beta$?
2. What does the solid line in Fig. 2d show? Isn't this a purely empirical plot?
3. How is allowing additional layers a "computational benefit" (l. 296ff)?

**Limitations:**

Limitations are discussed adequately. It could be mentioned that the experiments are limited in both the GNN architectures that were investigated as well as the datasets the experimetns were perfromed on.

The paper does not contain a broader impact statement or similar section on ethical considerations. This is perfectly acceptable for the work under review.

---

> ### Author Rebuttal · Authors · 2024-08-07
>
> We thank the reviewer for this thorough review and the helpful questions and
> comments. We will address them in the revised version:
>
> Re: Weaknesses
> 1. In the revision we will repeat the experiments shown in Fig. 3 on the Cora
> and/or CiteSeer datasets. For preliminary results on the Cora dataset,
> please refer to our global response.
> 2. While our formalism only captures GCN-like architectures, it still allows
> for freedom in designing a specific architecture: we
>     - allow for arbitrary activation functions (at the cost of evaluating Eq.
> (8) numerically),
>     - have a tunable parameter g allowing different weights on the diagonal
> of the shift operator (cf. Eq. (6)),
>     - and allow for bias.
>     - However, we agree that an extension of our formalism to other GNN
> architectures such as message passing ones would be interesting. We
> expect that such architectures should also exhibit a non-oversmoothing
> phase, although showing this is outside our current scope. We mention this as an interesting direction of future research in the discussion
> (cf. l. 341 ff).
> 3. We will improve this in the revision. Specifically we will state that we are
> linearizing Eq. (9), and that the factor $\frac{1}{2}(1+\delta_{\gamma,\delta})$ in Eq. (17) comes from
> the symmetry of the covariance matrix. In the revised version we will also
> point readers to appendix A, where the calculation is performed in full,
> immediately on l. 184. (Currently the reference to Appendix A appears
> only on l. 197.)
> 4. We will share the code upon acceptance.
>
> Re: Minor comments
> 1. We will add the following on l. 120: ”While the theory is agnostic to
> the choice of non-linearity, for our numerical experiments we use $\phi(x)=\text{erf}(\sqrt{\pi}/2 x)$; this choice allows us to carry out certain integrals analytically."
> 2. We will add the commas in the revised version to improve readability.
> 3. Indeed. We will correct this in the revision.
> 4. We will reduce this redundancy in the revision.
> 5. This is a valid suggestion: we have tried this and found that we can make
> it work without overloading the plot. The revised version therefore will
> have the labels directly in the figure.
>
> Re: Questions
> 1. No, it should be $K^{\text{eq}}_{\alpha\beta}$ instead of $K^{\text{eq}}$. We thank the reviewer for spotting this.
> 2. The solid line in Figure 2d) shows the zero line (we show this since the
> distances $d(x_\alpha, x_\beta)$ are bound by 0 from below). In the revised version,
> we will instead use 0 as the lower limit of the y-axis.
> 3. Indeed, the wording here might incorrectly suggest that adding layers
> would reduce computational cost. Rather we meant that adding more
> layers may make the network more expressive and/or improve its generalization error, since we see in Fig. 3(c,d) that the generalization error
> is minimal at around 15 layers. In the revised version we will write this
> as follows: “Thus by tuning the weight variance to avoid oversmoothing,
> we allow the constructions of GCNs with more layers and possibly better
> generalization performance.”

---

> > ### Comment · Reviewer_kVv2 · 2024-08-11
> >
> > Thank you for your response. It confirmed that this should clearly be accepted. I retain my score (already "Accept").
> >
> > In particular, seeing in the (preliminary) results on Cora that the method also works on typical benchmark datasets is good news for the technique. I only suggest adding a comment on the rough computational costs of the method in practical terms to the final paper. For example, something like: "Cora has X nodes and Y edges. For 10 layers, the computer Z took about n minutes/hours."

---

> > > ### Author Response · Authors · 2024-08-12
> > >
> > > This is a good suggestion, we will state the computational costs for our experiments on Cora in the final paper.

---

### Official Review · Reviewer_HWT8 · 2024-07-17

**Soundness:** 3
**Presentation:** 3
**Contribution:** 3
**Rating:** 5
**Confidence:** 3

**Summary:**

This work aims to understand whether GNNs in large depths always suffer from oversmoothing. The research starts from the equivalance of gaussian process (GP) and infinitely-wide NNs, and utilize eigenvalues of the linearization of GCN GP to quantify whether the model is in the phase of oversmoothing or not. After identify the intialization scale $$\sigma_w$$ as the key factor, it empirically verifies how deep GCN GPs and GCNs performance with different scales on synthetic graphs, and it turns out larger intialization does overcome the oversmoothing.

**Strengths:**

1. The motivation is natural and the result is good: oversmoothing is an important question in the community, and the initialization scale was not took care of much before.
2. The research path is clear: while it is hard to directly understand finite-width GNNs, the equivalence of GCN GPs and infinite-width GCNs  make it feasible to at least understand the problem in the wide regime, and then the argument could be empirically verified for finite widths.

**Weaknesses:**

1. Evaluation is a little weak for more practical settings: for example, add some experiments on classification tasks on simple datasets like cora, citeseer. The goal would be to provide a better measurement of the performance with large initialization, since the currently provided results on synthetic settings are less pesuasive for practitioners to use such large intialization vs other methods to overcome oversmoothing.

**Questions:**

1. a good addition would be to empirically verify how the eiganvalues change along training, which is to understand whether oversmoothing will appear or disappear with a certain scale of initialization.
2. what is the connection between this work and [1] about the singular values of weights? In section 3.3, they denote the singular values as $s$, and if the value is large, the bound for oversmoothing will not hold any more. Is this case the same as what is discussed in this work?

---

Reference:

[1] GRAPH NEURAL NETWORKS EXPONENTIALLY LOSE EXPRESSIVE POWER FOR NODE CLASSIFICATION.

**Limitations:**

Yes.

---

> ### Author Rebuttal · Authors · 2024-08-07
>
> We thank the reviewer for their review and helpful questions and comments. We will address them in the revised version:
>
> Re: Weaknesses
> 1.  In the revision we will repeat the experiments shown in Fig. 3 on the Cora and/or CiteSeer datasets. This should provide readers with a better sense of the generalization error they should expect in practice. For preliminary results on the Cora dataset, please refer to our global response.
>
> Re: Questions
> 1. Tracking the eigenvalues during training is an interesting proposal. We will provide such an empirical analysis in the revised version of the manuscript. We anticipate already some points from the presented material. For the wide networks we consider here, the Gaussian process is an accurate description of the posterior after training in a Bayesian setting. Such training could, for example, be performed with Langevin training, which is equivalent to gradient descent with weight decay and stochasticity. At large width weights change only marginally, so that we expect the GP at initialization to be an accurate description of the network also after training has converged. So in particular we expect that training will not change the eigenvalues in a notable manner in the limit of wide networks.
> Hence the condition for oversmoothing, derived from the GP at initialization, will be informative of the network’s oversmoothing properties also after training has converged.
> The more interesting setting, which we guess is the one envisioned by
> the referee, is that of networks of small or moderate width. Here due to
> training, weights typically depart from their values at initialization, which
> may change the eigenvalues after training converges if this is beneficial for
> minimizing the loss. Still in this regime we expect the obtained criterion
> for the start of the non-oversmoothing regime to be indicative of the network’s (over-)smoothing properties. We will check our expectations with
> numerical experiments in the revised version.
> 2. Indeed, the effect in [1] is similar to what we see in our work. However there are crucial differences between the two approaches. We will address the similarities and differences between our work and [1] more clearly in the revised version.
>     - The authors in [1] calculate an upper bound, which is loose for a large singular value $s$. This however does not imply that the networks dynamics are close to this bound, such that in their analysis GCNs with large s could still be oversmoothing (this happens e.g. in Figure 2 in [1] where the upper bound on distances between features increases, but in the simulation of the GCN distances decrease). In our approach on the other hand, we obtain not just bounds but quantitative predictions, both for the value of $\sigma_{w,\text{crit}}^2$ which defines the oversmoothing threshold, and for the expected feature distance given a particular $\sigma_w^2$. We were able to verify these predictions on finite-size networks (Fig. 1c, 2d, 3d).
>     - The authors in [1] find that their GNN architecture exponentially
> loses expressive power. In contrast, we determine a parameter setting
> where our GCN model does not lose expressive power exponentially:
> this happens precisely at the transition between the oversmoothing
> and non-oversmoothing regime that we discover here. At this point,
> there is one eigenvector $V^{(i)}_{\alpha\beta}$ in the sense of l. 191 f, for which the information propagation depth diverges, allowing information to propagate far into the network. Precisely at the transition, distances
> between feature vectors converge in a manner that is slower than
> exponential
>     - In [1] only the ReLU activation function is considered, while our
> formalism captures arbitrary non-linearities, at the cost of solving
> the integrals in Eq. (8) numerically.
>
> References:
>
> [1] Oono, Kenta, and Taiji Suzuki. "Graph neural networks exponentially
> lose expressive power for node classification." arXiv preprint arXiv:1905.10947
> (2019).

---

### Author Rebuttal · Authors · 2024-08-07

We thank the reviewers for their time and valuable comments. A common point
across reviews was that the paper would be strengthened by adding results
which use a real world dataset instead of just the synthetic CSBM (contextual
stochastic block model). We agree with this assessment, so we will address it
in the revision. Our preliminary results using the Cora dataset show the same
qualitative behaviour as with our CSBM example: Beyond the transition to
chaos the network no longer oversmoothes and achieves low generalization error
for many layers. In the attached pdf you can find the figure corresponding to our Fig. 3a
for the Cora dataset.

Replies to more specific comments are included in the author-specific responses.

---

### Decision · Program_Chairs · 2024-09-25

**Decision:**

Accept (poster)

**Comment:**

All reviewers unanimously voted for acceptance.

A common point across reviews was that the paper would be strengthened by extending the empirical analysis to real world dataset. As a consequence authors presented experiments on Cora during the rebuttal, which I encourage the authors to incorporate in the final version of their paper.